# Emerging Highly Virulent Porcine Epidemic Diarrhea Virus: Molecular Mechanisms of Attenuation and Rational Design of Live Attenuated Vaccines

**DOI:** 10.3390/ijms20215478

**Published:** 2019-11-04

**Authors:** Yixuan Hou, Qiuhong Wang

**Affiliations:** Food Animal Health Research Program, Ohio Agricultural Research and Development Center, College of Food, Agriculture and Environmental Sciences, Department of Veterinary Preventive Medicine, College of Veterinary Medicine, The Ohio State University, Wooster, OH 44691, USA; hou.214@buckeyemail.osu.edu

**Keywords:** porcine epidemic diarrhea virus, coronavirus, virulence, attenuation, live attenuated vaccine

## Abstract

The highly virulent porcine epidemic diarrhea virus (PEDV) emerged in China in 2010. It infects pigs of all ages, and causes severe diarrhea and high mortality rates in newborn pigs, leading to devastating economic losses in the pork industry worldwide. Effective and safe vaccines against highly virulent PEDV strains are still unavailable, hampering the further prevention, control and eradication of the disease in herds. Vaccination of pregnant sows with live attenuated vaccines (LAVs) is the most effective strategy to induce lactogenic immunity in the sows, which provides A passive protection of suckling piglets against PEDV via the colostrum (beestings, or first milk) and milk. Several LAV candidates have been developed via serially passaging the highly virulent PEDV isolates in non-porcine Vero cells. However, their efficacies in the induction of sufficient protection against virulent PEDV challenge vary in vivo. In this review, we summarize the current knowledge of the virulence-related mutations of PEDV and their potential roles in PEDV attenuation in vivo. With the successful development of reverse genetics systems for PEDV, we also discuss how to use them to generate promising LAV candidates that are safe, effective and genetically stable. This article provides timely insight into the rational design of effective and safe PEDV LAV candidates.

## 1. Introduction

Porcine epidemic diarrhea virus (PEDV) infects pigs of all ages and causes acute diarrhea, vomiting, anorexia, weight loss, dehydration and even death [1,2,3]. PEDV belongs to the genus *Alphacoronavirus* within the family *Coronaviridae* [4]. It is an enveloped virus with a 28 kb single-stranded, positive-sense RNA genome. The genome contains 5’-cap structures (an N7-methyl guanosine and a methylation on the 2’O position of the first nucleotide), a 3’-poly(A) tail, and six known open reading frames (ORFs), namely nonstructural protein genes ORF1a/1b, four structural protein genes [spike (S), membrane (M), envelope (E), and nucleocapsid (N)], and an accessory protein gene ORF3. The ORF1a/1b encodes two polypeptides (pp1a and pp1ab) that can be processed into 16 nonstructural proteins (nsps) (Figure 1). The expression of pp1ab is mediated by a −1 frameshifting signal UUUAAAC near the end of ORF1a [5]. Upon translation, a papain-like protease (PLpro) domain within the nsp3 cleaves the junctions and releases nsp1 and nsp2 from the polypeptides. Subsequently, a 3C-like protease cleaves the junctions between nsps 3 to 16 [6]. 

These nsps are responsible for critical steps in the PEDV life cycle, including genomic RNA replication, sub-genomic messenger (sgm) RNA synthesis and interaction with various host factors. The glycoprotein S binds to host receptors and triggers virus-host membrane fusion during virus entry. These two steps are mediated by the two functional subunits of the S protein, S1 and S2, respectively. Currently, the protein receptor for PEDV entry remains unknown [7,8]. The structural proteins M and E mainly mediate the assembly of enveloped viral particles [9]. The N protein is a multifunctional viral protein involved in multiple steps in viral replication and regulating host functions [10,11,12,13]. The accessory protein ORF3 is identified as an ion channel and has multiple regulatory functions [14,15,16]. 

The first PEDV case was reported on a swine farm in the UK in 1971 [17]. Since then, this virus has been identified throughout different countries in Europe and Asia [18]. These PEDV strains are classified as the classical strains or genotype 1a (G1a) strains based on the S genes/proteins [19]. Due to the fact that G1a PEDV strains cause moderate mortality in neonatal piglets and the increased biosecurity practices on farms in Europe, PEDV had limited economic impacts, and no PEDV vaccines have been developed in Europe [20]. On the other hand, G1a PEDV had caused significant economic losses in Asian countries from the 1980s to the 2000s. Vaccines were developed and applied in the field, including several live attenuated vaccines (LAVs) that were generated by passaging virulent G1a PEDV isolates in Vero cells, a simian kidney epithelial cell line that is deficient in type I interferon production but expresses interferon receptors [21]. In Japan, P-5V was employed as a commercial intramuscular (IM) LAV since 1997 [22,23]. In China, a bivalent LAV composed of the PEDV CV777 strain and the TGEV H strain was licensed in 1999 [24]. Subsequently, a trivalent LAV containing these two viruses and a porcine rotavirus strain NX was licensed in China in 2015 [25]. In South Korea, LAVs using PEDV strains KPEDV-9 and DR13 were licensed in 1999 and 2004, respectively [26,27].

However, highly virulent PEDV variants causing up to 100% mortality in neonatal piglets suddenly emerged in China in 2010 and spread quickly to many Asian countries [1,3]. Many G1a LAV-vaccinated swine farms still suffered from these highly virulent PEDV outbreaks [1,25], suggesting that the efficacy of G1a LAVs against the emerging PEDV strains was minimal, and that PEDV vaccines based on those new strains are needed. 

It was not until spring 2013 that the highly virulent PEDV was introduced into the United States (US) and caused epidemic outbreaks in PEDV-naïve swine population [2], leading to the death of 10% (~seven million) US pigs, and economic losses of $900 million to $1.8 billion from 2013 to 2014 [28,29]. Phylogenetic analyses of viral genomes separated the classical and the emerging highly virulent PEDV strains into two distinct branches. The latter are classified as G2 or non-S insertion and deletion (INDEL) PEDV [19,25,30]. The US G2 PEDV is also named as the “US PEDV prototype” [31] or “original US PEDV” [32] by different research groups. In addition to the G1a and G2 strains, several other variants have been discovered. For example, two new PEDV mutants with large variations in the S gene have been reported in different countries: (1) S INDEL or G1b PEDV strains, which are classified based on the S genes/proteins, are natural recombinant PEDVs with a G2-like genomic backbone carrying an S1 region of G1a strains [33,34]; and (2) S1 N-terminal domain-deletion (NTD-del) strains that are G2-like strains containing a 194 to 216-aa deletion within the N-terminal domain of the S1 subunit [35,36,37,38]. Compared with highly virulent G2 strains, both S INDEL and S1 NTD-del strains are less virulent in pigs [31,32,39,40,41]. These PEDV variants often co-circulate on pig farms, increasing the potential for the emergence of new recombinants due to the frequent recombination events among coronaviruses (CoVs) [34,35,36,42]. To date, PEDV outbreaks have not been documented in Africa and Oceania. In Asia, all of the PEDV genotypes and variants (G1a, G1b, G2 and S1-NTD-del) have been reported, whereas no G1a strain has been identified in the Americas [30,34]. The G1a and G1b strains are the main PEDV strains circulating in European countries, with the only reported G2 PEDV outbreak occurring in the Ukraine [43,44]. Currently, PEDV G2 strains are predominant in pig farms in Asian and American countries [25,34,45,46]. These different epidemiological patterns suggest the complicity of viral evolution (point mutations, insertions, deletions and recombination) in herds where multiple PEDV genotypes/variants are co-circulating and/or vaccines with varied efficiency are applied. 

Among all the PEDV genotypes, the G2 highly virulent strains caused the most severe clinical signs, leading to 50–100% mortality rates in neonatal piglets [1,2,47,48]. Because neonatal pigs do not have enough time to develop anti-viral immune responses before they encounter the deadly virus, the vaccine targets are mainly pregnant sows that can be immunized and generate lactogenic immunity to passively protect suckling piglets via colostrum and milk. Currently, effective and safe PEDV vaccines against G2 strains are still not available, although several vaccines have been developed. For example, one Venezuelan equine encephalitis virus vectored subunit (S protein) vaccine and an inactivated whole-virus vaccine based on G2 PEDV strains have been commercially available in the USA since 2014 [49]. However, the efficacy of both vaccines in the protection of suckling pigs from disease and death are questionable [49]. A potential reason would be that these vaccines administered intramuscularly do not replicate within the intestines. Early studies of another swine enteric alphacoronavirus, the transmissible gastroenteritis virus (TGEV), discovered that newborn piglets can be protected from the viral infection only through ingesting virus-specific antibodies in the colostrum and milk [50,51] of sows orally inoculated with the live TGEV, but not parenterally immunized with inactivated or subunit vaccines [52,53,54,55]. Therefore, orally priming seronegative sows with effective LAVs is a promising vaccination strategy against PEDV G2 strains. 

In this review, we will summarize the virulence-related mutations in G2 PEDV strains. Such information is valuable, and can be applied to the rational design of LAV candidates to prevent and control porcine epidemic diarrhea (PED), the most deadly enteric viral disease of suckling pigs.

## 2. Mutations in Vero Cell-Attenuated G2 PEDV Strains 

To date, several Vero cell-attenuated G2 strains have been reported, including US isolates PC22A [56] and 8aa [57], as well as Asian isolates YN [58], Pingtung-52 [59], KNU-141112 [60] and Zhejiang08 [61]. In this section, we focus on four attenuated G2 PEDV strains that share common patterns of mutations in their genomes. Table 1 summarizes all accumulated amino acid-mutations in the genomes of four attenuated PEDV G2 strains, PC22A-P120 (the 120th passage), YN144 (the 144th passage), PT-P96 (the 96th passage), and KNU-141112 DEL5/ORF3 (the 100th passage), compared with their corresponding virulent parental viruses. 

Among these attenuated PEDVs, the S proteins contain the predominant mutations compared with other viral genes, and the S2 subunit harbors more amino acid-substitutions than S1. This phenomenon is perhaps due to the adaptation of these PEDV to the non-porcine receptors on Vero cells and to the trypsin cleavage in vitro. For instance, mutations can be found in the sialic acid-binding domain S0 of PC22A-P120, PT-P96 and KNU-141112 DEL/ORF3. The PT-P96 even acquired an F554S mutation within the projected receptor binding domain, the main VN epitope COE. Mutations which accumulated in the S2 ectodomain of these Vero-adapted PEDV strains may be associated with protease cleavage and fusion. A Q893K mutation in PC22A-P120 is located at two amino acids upstream of an identified trypsin cleavage site next to the fusion peptide [62]. These mutations in the S protein may impair the recognition by the porcine receptor, contributing to the inefficient replication of these PEDVs in vivo. 

The common mutations among these attenuated PEDVs suggest universal mechanisms in the adaption to Vero cells. For example, both YN144 and KNU-141112 DEL5/ORF3 contain deletions in their ORF3 gene, which also commonly exists in many Vero cell-adapted G1 PEDV strains, such as attenuated CV777 and DR13 [63,64]. One possibility is that PEDV loses its dispensable interferon antagonist gene ORF3 after continuously adapting to interferon-deficient Vero cells [21]. We also found that all of the four attenuated PEDVs contain mutations in residue #1564 and/or #1565 of nsp3. Regardless the virulent or attenuated PEDV, residue #1564 could be serine (S) or phenylalanine (F). Moreover, premature stop codons exist in the cytoplasmic tail of the S protein in both PC22A-P120 and KNU-141112 DEL5/ORF3, resulting in a partial deletion of an endocytosis signal YxxΦ and the complete deletion of an endoplasmic reticulum (ER)-retrieval signal KVHVQ. We demonstrated previously that the loss of these two motifs results in enhanced viral syncytia formation in Vero cells and attenuation in pigs [65]. Finally, both YN144 and PT-P96 harbor a C1354F substitution in the cytoplasmic tail of the S protein, suggesting another common alteration involved in S intracellular sorting or virion assembly. Collectively, these features may represent common virulence-related mutations which can be applied in the rational design of PEDV LAV candidates.

Protective efficacies of these reported G2 PEDV LAV candidates have not been thoroughly evaluated in vivo. Ideally, an efficacious PEDV LAV should induce protective lactogenic immune responses in sows. Since experiments of sows are expensive and labor-intensive, many studies used neonatal or weaned pigs as models to evaluate the degree of attenuation and immune responses induced by these LAV candidates. However, some of these studies lacked virulent PEDV-challenge data. The potential efficacy of KNU-141112 DEL5/ORF3 and Zhejiang08 were instead extrapolated from serum VN antibody titers and dendritic cell activation in vaccinated neonatal piglets, respectively [60,61]. In contrast, the efficacy of PC22A-P120 and PT-P96 were evaluated by challenging the vaccinated neonatal piglets with homologous virulent strains at three and four weeks post-inoculation, respectively [59,66]. Their protective efficacy was quantified by the severity of clinical signs, morbidity and mortality, fecal consistency scores, fecal PEDV shedding and humoral antibody titers in the challenged pigs. The study of PC22A also compared protective efficacies of the Vero cell-attenuated PEDVs at different passage levels, including P120 and P160. However, the results showed that the protective efficacy correlates negatively with the degree of attenuation of the virus in piglets. In a subsequent study of weaned pigs, the fully attenuated PC22A-P120 induced insufficient mucosal immune responses, which may be the main reason for the low protection rate [67]. 

The low protection rates of the Vero cell-attenuated LAV candidates are potentially attributed to several reasons. It is possible that the abundant mutations accumulated in the S genes compromised the replication and immunogenicity of these LAV candidates in pigs (Table 1). Instead of using Vero cells, attenuating PEDV isolates using swine cell lines may decrease the number of mutations in the S protein. Further experiments are needed to prove this hypothesis. Nevertheless, these results suggest that a better version of PEDV LAV candidates should contain the critical epitopes in the S protein that are conserved in the wild type (WT) viruses, to retain immunogenicity. 

## 3. Reverse Genetics Systems and Their Applications in the Studies of the Molecular Mechanisms of PEDV Attenuation

To date, several PEDV reverse genetics systems have been developed. Teeravechyan et al. [68] thoroughly reviewed the principles, advantages and drawbacks of three different approaches using (1) targeted RNA recombination [69], (2) bacterial artificial chromosome (BAC) [70], and (3) in vitro ligation methods. Recombinant PEDVs generated using reverse genetics technology can be attenuated and retain sufficient immunogenicity. In general, it is more challenging to rescue recombinant PEDVs from the genomes of virulent PEDV strains than that from cell culture-adapted PEDV strains. The spread of progeny viruses to adjacent cells seems critical to the successful rescue of recombinant viruses in cell culture. Many virulent PEDV strains trigger moderate syncytia in Vero cells, and require supplementation of trypsin in the culture medium [62,65]. 

The inefficient replication and inability to induce cytopathic effects (CPE) initially hampered the rescue of recombinant PEDVs from the infectious cDNA clones of two US original highly virulent strains (PC22A [71] and Colorado/2013 [72]) in Vero cells. Although no syncytia occurred, the supernatants of the recombinant PC22A-transfected cells contained sufficient viral particles to infect neonatal piglets [71]. This reverse genetics system was improved subsequently by introducing a tri-nucleotide insertion (GGC) into the S protein [39]. The insertion resulted in the replacement of an aspartic acid by a glycine residue and the introduction of an additional histidine residue (D466GH). These changes allowed the recombinant PEDV, (the infectious clone-derived PC22A (icPC22A)), to induce syncytia and be easily rescued and plaque-purified in cell culture. Later, Deng et al. optimized the infectious clone of the Colorado/2013 strain using the same strategy [72]. Another example for the enhanced viral spreading is the BAC vector infectious clone of a Thai G1a strain AVCT12: The infectious PEDV could be rescued only if the two intracellular signaling motifs (YxxΦ and KVHVQ) at the cytoplasmic tail of the S protein and the ORF3 were ablated [73]. Later, we found that the YxxΦ motif triggers endocytosis of S proteins, the motif KVHVQ is involved in retention of the S proteins in the ER-Golgi intermediate compartment and the loss of both motifs significantly enhanced syncytia formation in Vero cells and reduced virulence in pigs [65]. In addition to these mutations in the S gene, improving the susceptibility of the cell line to PEDV infection would also enhance the recovery of infectious viruses from the cDNA clone. For example, the recovery of the recombinant AVCT12 also relied on a Vero cell line that stably expressed porcine aminopeptidase N (pAPN) [73]. Although pAPN is not the cellular receptor for PEDV, its peptidase activity facilitates the entry of PEDV into cells [7,8,74,75]. 

A PEDV infectious cDNA clone based on a highly virulent strain is a useful tool to study the molecular attenuation mechanisms of PEDV. One good example is the identification of the function of the S0 domain in PEDV virulence. The S0 domain is the most N-terminal domain in the S protein, and contains about 230 amino acid residues. Several PEDV variants with a large deletion (194 to 216 aa) in the S0 domain have been identified in clinical samples [35,36,37]. One field strain, TTR-2, was attenuated in neonatal piglets. However, whether the large deletion was the genetic determinant for attenuation is unclear, because other mutations were also identified in the genome [38,41]. The direct evidence for the attenuation effect of this large deletion was confirmed using a recombinant PEDV icPC22A-∆197 that is different from the highly virulent icPC22A only in the 197-aa deletion [39]. Virulent PEDV infectious cDNA clones have also been applied to study the role of S1 subunits of different PEDV genotypes. Two studies generated chimeric PEDVs with the genomic backbones of G1 and G2 strains, but with the S1 subunits from different genotypes [76,77]. Their results suggest that the S1 subunit contributes to the virulence differences observed between G1 and G2 PEDV strains, but is not the only genetic determinant. 

## 4. Rational Design of LAV Candidates Using Reverse Genetics Technology

The efficacy of oral LAVs correlates with the replication of PEDV in the swine intestines. However, increasing evidences have shown that Vero cell-attenuated PEDVs replicate poorly in pigs. Therefore, to maintain the relatively efficient replication of PEDV in pigs, mutations in the S protein must be minimized without compromising attenuation. To date, a few studies generated PEDV LAV candidates using reverse genetics technology and evaluated the pathogenicity and immunogenicity in pigs. First, we generated a recombinant PEDV icPC22A-KDKE4A via abolishing the 2’-O methyltransferase (MTase) activity of the nsp16 of PC22A strain [78]. This viral enzyme is responsible for the methylation of the 2’-O in the first ribose of viral RNA. Previous studies demonstrated that the inactivation of the 2’-O MTase sufficiently attenuates some betacoronaviruses, including mouse hepatitis virus (MHV) [79], acute respiratory syndrome (SARS)-CoV [80] and Middle Eastern respiratory syndrome (MERS)-CoV [81,82] in mice. Next, we further inactivated the endocytosis signal YxxΦ motif in the cytoplasmic tail of the S protein to generate icPC22A-KDKE4A-SYA. We found that the two PEDV mutants induced earlier and stronger type I and type III interferon responses in vitro. 

After oral administration, both icPC22A-KDKE4A and icPC22A-KDKE4A-SYA caused minimal clinical signs and no death in neonatal gnotobiotic piglets, but triggered sufficient protection post challenge three weeks later. In another study, Deng et al. exploited the knowledge of CoV endoribonuclease nsp15 [83,84] to generate a recombinant PEDV icPEDV-EnUmt that carries an inactivated nsp15. This virus lacked the function of regulating the dsRNA level and antagonized the type I and type III interferon responses [72]. This PEDV mutant was attenuated in neonatal piglets and triggered earlier and stronger innate immune responses in cell culture. However, no challenge study was performed. Most recently, Pascual-Iglesias and colleagues generated a TGEV-PEDV chimeric virus rTGEV-RS-SPEDV by replacing the ectodomain of the S protein of an attenuated TGEV with that of a G2 PEDV strain. The genomic backbone of the TGEV harbors duplicated transcriptional regulatory sequences (TRSs) of M, N and ORF7 genes, which lead to attenuation in pigs. This recombinant virus was partially attenuated in 5-day-old piglets, and induced partial protection in 21-day-old pigs from a highly virulent PEDV challenge [85]. These studies utilized newborn or young pigs to evaluate the attenuation and immunogenicity of the recombinant PEDVs. Further experiments are needed to validate the efficacies of these LAVs candidates in pregnant sows.

Efforts to attenuate other CoVs using reverse genetics have focused on modifying a variety of viral genes with non-essential functions, including nsp-coding sequences, E and accessory genes. ORF3 is the only known accessory gene, and is dispensable for PEDV replication [4,6]. However, the removal of this ORF3 gene only partially attenuates a highly virulent PEDV in piglets [71], suggesting the necessity of introducing additional modifications to generate LAV candidates. Table 2 summarizes the reported recombinant CoVs with modifications in virulence-related genes (except for S and accessory genes). These modifications targeted CoV IFN antagonism and other non-essential functions for replication. Because most of these functional motifs are conserved among CoVs, some of these strategies can also be applied to the design of PEDV LAV candidates. In addition to the nsp15 and nsp16, inactivation of the IFN-antagonistic function of other viral genes could also attenuate a CoV in vivo. For instance, abolishing the anti-IFN function of the nsp1 of MHV and SARS-CoV attenuated these viruses in mice [86,87]. For PEDV, it has been reported that at least 11 viral proteins suppress IFN responses, including nsp1, nsp3, nsp5, nsp7, nsp14, nsp15, nsp16, ORF3, E, M and N. Although these IFN-antagonistic proteins regulate different steps in the IFN pathways, inactivation of each of them may lead to attenuation of PEDV in pigs. Recombinant PEDVs lacking these IFN suppressing functions may induce an earlier onset of and enhanced IFN responses in host cells, leading to improved immune responses compared with the WT virus. In addition to the accessory gene ORF3, several nsps are dispensable for CoV replication. Deletion or inactivation of these nsps causes the attenuation of different CoVs in vivo. A previous study demonstrated that recombinant MHV and SARS-CoV lacking the entire nsp2 were still viable in vitro, although their phenotypes in vivo remained undefined [88]. Many studies of CoV LAVs focused on the inactivation of non-essential enzymatic functions, including the RNA transcription-regulatory function of nsp1 and the exoribonuclease activity of nsp14 [86,87,89]. Another well-characterized approach for CoV LAV development is to delete the structural protein E. Since the E protein is critical for CoV virion assembly, engineered cell lines that continuously express the E protein were used to support the replication of these CoV mutants in vitro. It has been shown that SARS-CoV and MERS-CoV lacking the entire E protein were attenuated in mice and induced protective immunity [90,91]. However, the attenuated SARS-CoV lacking the full-length E gene reverted to a virulent phenotype after serial passages in vivo due to the incorporation of a new chimeric protein with a PDZ-binding motif mimicking that of the E protein [86], suggesting that the deletion of an entire gene may not be an effective strategy to maintain the genetic stability of a LAV candidate. 

A major safety issue of applying LAVs in the field is the reversion to virulence caused by mutations and RNA recombination. Efforts have been made to improve the stability of the introduced mutations to prevent reversion. One strategy is to introduce multiple virulence-related mutations into different genes of the viral genome. The above introduced recombinant PEDV icPC22A-KDKE4A-SYA retained the introduced mutations in 2-O’ MTase and S protein after passaging in pigs three times, indicating its genetic stability in vivo [78]. The same approach has been demonstrated for other recombinant CoVs. A recombinant SARS-CoV resistant to reversion was generated via deleting both nsp1 and E proteins [86]. A similar effect was also observed in another recombinant SARS-CoV with an inactivated exonuclease domain in nsp14 and inactivated 2’-O MTase function in nsp16 [82]. 

In summary, knowledge from PEDV and other CoV studies help us develop a systematic strategy for the rational design of PEDV LAV candidates: 1) Target the genes that are not essential for viral replication and immunogenicity, such as the genes encoding modulators of innate immune responses and virus replication, non-VN epitopes of structural protein S, and the accessory gene ORF3; 2) Make a panel of mutations that contains small deletions/mutations for each targeted gene instead of deleting entire genes to increase genetic stability; 3) Introduce at least two distinct mutations into separate genes that attenuate the virus to further increase genetic stability. By selecting for at least two distinct mutations across the 28 kb-genome of PEDV, vaccine candidates are more resistant to mutation-driven reversion. In addition, double or more recombination events would be required to replace all of the attenuated gene copies with WT copies. Also, to improve the stability of PEDV LAVs, another strategy is to re-design the TRS in the PEDV genome to prevent recombination. Recombinant SARS-CoVs bearing the re-designed TRS partially resisted homologous RNA recombination in a co-infection event [99,100]. 

## 5. Conclusions

Highly virulent G2 PEDV outbreaks have caused immense economic losses in the pork industry and are still causing epidemic and endemic outbreaks in many countries. However, effective and safe vaccines are not commercially available. Reverse genetics technology provides a very useful platform in studying the virulence-related mutations and in the rational design of PEDV LAV candidates. As a member of the Coronaviridae family, PEDV shares many similar gene functions with other CoVs. Although current understanding of PEDV biology is still limited, knowledge from other CoVs may aid in the identification of mutagenesis targets and help achieve the optimal balance between attenuation and immunogenicity. An ideal PEDV LAV should replicate effectively in pig intestines, not cause diseases, induce enough protective immune responses and not revert to a virulent phenotype. In the future, rationally designed PEDV LAV candidates bearing different genetic modifications should be evaluated in pregnant sows that are the major targets of PEDV vaccination, and can passively protect suckling piglets from PEDV disease via the PEDV-specific neutralizing antibodies in colostrum and milk. 

## Figures and Tables

**Figure 1 ijms-20-05478-f001:**
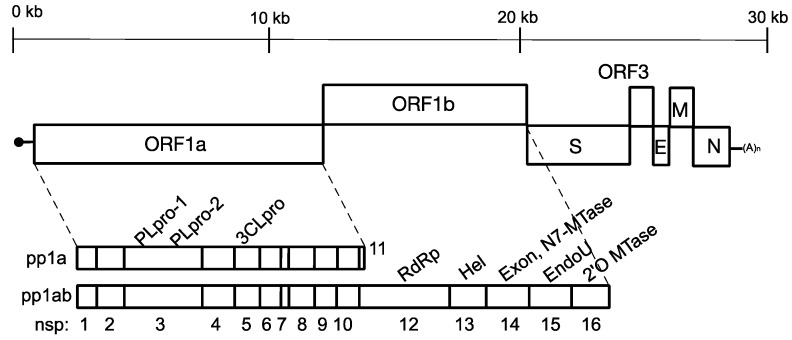
Schematic organization of the porcine epidemic diarrhea virus (PEDV) genome, polypeptides pp1a and pp1ab. The genome encodes open reading frames 1a (ORF1a) and 1b (ORF1b), followed by the genes encoding spike protein (S), accessory protein 3 (ORF3), envelope protein (E), membrane (M) and nucleocapsid (N) proteins. Translation of ORF1a and 1b results in two polypeptides pp1a and pp1ab, mediated by a −1 frame-shifting signal [5]. The polypeptides are protease-processed into 16 non-structural proteins (nsps). PLpro: Papain-like protease; 3CLpro: 3C-like protease; RdRp: RNA-dependent RNA polymerase; Hel: Helicase; Exon: Exonuclease; MTase: Methyltransferase: EndoU: Endoribonuclease.

**Table 1 ijms-20-05478-t001:** Changes of amino acids among four Vero cell-attenuated PEDV G2 strains.

Genes		PC22A-P120	YN144	PT-P96	KNU-141112 DEL5/ORF3
Nsp2			P509S	K159N, T510I	
Nsp3		S1564F, I1565F, D1622G	I1565T	F1564S	S1564F
Nsp4			H2925Y	E2937A	
Nsp5					T3186I
Nsp6			V3505E		
Nsp7		A3627V			
Nsp12		L4622F	V4832F		
Nsp14			L6136F, V6137S		
Nsp13					T5132I, A5272S
Nsp15				M6393I	
S1	domain 0	∆55 to 56 E57K, I166V		T144I	K272T
	domain A	Q454D, D466G, ^477H	D405G, D428A, R490T		T383N
	domain B/COE			F554S	
	domain C & D		S722R		
S2	ectodomain	V811F, Q893K, A971V, G1009V, F1015L	T779N, Q825H, S968A, H1045Q, D1165N, F1210Y, S1218G, I1304L	S887R, S968A, I1021S, R1026K, L1252R	G888R, E1287Q
	cytoplasmic tail	E1379 stop	C1354F	C1354F, C1358F	F1380H, E1380D, 1381stop
ORF3		I98T	138 to141 YYDG to FMTA, 142 to 145 KSIV to NPL.	Y170H	∆26D, N166S
E					P70S
M		L255I	I12V, S79A, F145L		

1. The passage numbers of the Vero cell-attenuated PEDVs are listed after each strain name: PC22A-P120, 120th; YN144, 144th; PT-P96, 96th; KNU-141112 Del5/ORF3, 100th. 2. Domain 0, residues 19 to 230; domain A, residues 231 to 498; domain B/COE, residues 199 to 638; domain C & D, residues 639 to 729; S2 subunit, residues 760 to 1387, fusion peptide, residues 895-901; cytoplasmic tail, residues 1350 to 1387. 3. The residue number of domains is based on the S protein of PC22A strain. 4. ∆55 to 56, deletion of residues 55 to 56; ^477H, insertion of a H at residue 477; ∆26D, deletion of a D at residue 26. 5. Locations of mutations in ORF1a/1b are shown in the residue number in polypeptide 1ab.

**Table 2 ijms-20-05478-t002:** Summary of recombinant coronaviruses (CoVs) with virulence-associated modifications (without S and ORF3 proteins).

Modified Viral Protein.	Virus	Introduced Mutation	Inactivated Function	Phenotypes in Vitro	Evaluation of Attenuation in Vivo	Induction of Protection against the Challenge	Reference
Nsp1	MHV	Deletion of 99 aa (residue: 829927–)	IFN antagonism	The mutant MHV replicated normally in vitro and induced type I IFN in antigen-presenting cells.	Yes	Complete protection against homologous virulent strain	[92]
	MHV	Deletion of conserve motif LLRKxGxKG (residue: 191199–)	Regulation of host gene expression	The mutant MHV replicated slightly slower than WT virus.	Yes	Complete protection against homologous virulent strain	[87]
	SARS-CoV	Deletion of conserve motif LLRKxGxKG (residue: 121129–) or motif D (residue 154 to 164)	Regulation of host gene expression or other	The mutant viruses replicated similarly to WT virus.	Yes	A rSARS-CoV with both ∆nsp1 (D motif) and ∆E provide complete protection against homologous challenge	[86]
Nsp2	MHV and SARS-CoV	Deletion of entire nsp2	Unknown	The mutant viruses replicated less effectively and had decreased viral RNA synthesis compared with WT virus.	No	N/A	[88]
Nsp3	MHV	V787S	Ubiquitin-like domain in papain-like protease	Mutant replicated efficiently but its protease activity was reduced.	Yes	Complete protection against homologous virulent strain	[93]
	MHV or SARS-CoV	N1347A (MHV); N1040A (SARS-CoV)	ADP-ribose-1’-phosphatase	The mutant viruses replicated similarly to WT virus.	Yes	N/A	[94,95]
Nsp5	MHV	T26I/D65G	3C-like protease inhibitor-resistant mutations	The MHV mutant resisted to a 3C-like protease inhibitor and replicated less effectively compared with WT virus in vitro.	Yes	N/A	[96]
Nsp14	SARS-CoV	D90A/E92A	Catalytic motif I of the exonuclease	The SARS-CoV mutants had impaired proof-reading function.	Yes	Complete protection against homologous virulent strain	[89]
	TGEV	H157C	Zinc finger 1 of the exonuclease	Accumulation of dsRNA in the infected cells at late stage of infection.	No	N/A	[97]
Nsp15	MHV or PEDV	H262A (MHV) or H226A (PEDV)	Endonuclease	The mutants induced early and robust IFN responses.	Yes	MHV H262A induced complete protection.	[72,83]
Nsp16	MHV, SARS-CoV, MERS-CoV, PEDV	D129A (MHV), D130A (SARS-CoV and MERS-CoV) or KDKE to AAAA (PEDV)	Catalytic tetrad of 2′-O methyltransferase	The mutants induced early and robust IFN responses	Yes	Induction of protection against virulent virus challenge	[78,80,81]
E	SARS-CoV or MERS-CoV	Deletion of entire E protein	Assembly of virions	Mutant CoVs replicated in cells expressing E protein.	Yes	Induction of protection against virulent virus challenge	[91,98]

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
