# Peer review of "Emerging Highly Virulent Porcine Epidemic Diarrhea Virus: Molecular Mechanisms of Attenuation and Rational Design of Live Attenuated Vaccines"

_ijms, 2019, doi:10.3390/ijms20215478_

Round 1
Reviewer 1 Report
In this review article, Hou and Wang first briefly review the history and diverse genotypes of PEDV, and the classical approach of developing PED vaccines. They summarize the current knowledge of the virulence-associated mutations within coronavirus genomes. With the successful development of reverse genetics systems for PEDV, they further discuss potential approaches to generate live-attenuated vaccine candidates. Overall, this is a well-crafted manuscript. The information summarized in this article not only provides broad and detailed information about coronavirus pathogenesis and virulence-determinants but also would guide the rational design of effective vaccine candidates.
Minor comments for authors’ consideration:
1) Attenuating PEDV pathogenicity by serially passing it in Vero cells was successful for developing vaccines against the classical PEDV strain. In this article, the authors discuss the disadvantages of this approach. Do the authors have any comments why this approach works for the classical strain but doesn’t work for current endemic strain?
2) Table 1 has some notes on the table. These notes should be separated from the table.
Author Response
Minor comments for authors’ consideration:
1) Attenuating PEDV pathogenicity by serially passing it in Vero cells was successful for developing vaccines against the classical PEDV strain. In this article, the authors discuss the disadvantages of this approach. Do the authors have any comments why this approach works for the classical strain but doesn’t work for current endemic strain?
Response: It took about two decades to develop the classical PEDV LAV vaccines in China, Japan and Korea. The generation of LAVs using Vero cell adaptation is trial and error. Therefore, effective LAVs may also be generated for the emerging highly virulent PEDV if people keep trying to adapt different strains in Vero cells. However, it would be faster if we generate a LAV using reverse genetics after we know which genes can be modified to attenuate a PEDV and retain its immunogenicity.
2) Table 1 has some notes on the table. These notes should be separated from the table.
Response: We have moved all the footnotes out of the table.
Reviewer 2 Report
This review summarizes the different high virulent strains of PEDV and the rational design of live attenuated vaccines. This manuscript has been structured according to the historical methodology used in the generation of LAVs, however, there is a lot of redundancies among sections. This manuscript should benefit from extensive reorganization to avoid repetitive discussions.
Figure 1 provides a very good summary of the genome organization. Authors should make sure that all the terms referred to as in the text are depicted in Figure 1 with the relevant abbreviations. Also a legend should be written to describe the schematic.
Related to the previous point, in line 30, the sentence suggests that there are multiple cap structures at the 5’end. This should be clarified.
When describing the two polypeptides (pp1a and pp1ab) the authors should indicate how they are generated.
In line 41, there is an extra space after “ion channel and ..”
In line 50, it should read “live attenuated vaccines”
In Line 52, introduction of Vero cells should be accompanied by a description that defines it as a cell line incapable of secreting IFN but that still are expressing IFN receptors
In line 90, the author should provide a brief description explaining why the vaccines that have been commercially available in the USA were not as efficacious.
Line 110: are the authors referring PC22A-P120, PT-96 and KNU-141112 DEL/ORF3 as G2 strains. This should be fixed in the text.
Table 1 is hard to follow. The authors should add footnotes for clarity and should be consistent in the nomenclature of the different mutants.
In line 124: ER should be referred as endoplasmic reticulum and later in the text should be referred as ER.
In line 137, attenuated virus Zhejian08 has not been described in table 1.
The text of this section: Mutations in Vero cells-attenuated G2 PEDV strains would benefit from separating paragraphs to help readers.
The text should include the specific abbreviations for the different amino acids
The following statement: “the higher the passages in Vero cells, the less replication of the virus is, leading to milder diseases and weaker immune responses in neonatal or weaned pigs [63,64]. One potential reason is the excessive adaptation of PEDV to the simian cell line resulted in accumulated mutations in the S protein” has already being discussed before. The authors should considered reorganizing the sections to avoid overlapping and redundancies.
The topic of recombination should be addressed when discussing viral evolution (line 79)
Author Response
This review summarizes the different high virulent strains of PEDV and the rational design of live attenuated vaccines. This manuscript has been structured according to the historical methodology used in the generation of LAVs, however, there is a lot of redundancies among sections. This manuscript should benefit from extensive reorganization to avoid repetitive discussions.
Response: We have removed the redundant sentences.
Figure 1 provides a very good summary of the genome organization. Authors should make sure that all the terms referred to as in the text are depicted in Figure 1 with the relevant abbreviations. Also a legend should be written to describe the schematic.
Response: Thank you for the suggestion. We have added a detailed legend to briefly describe the genome organization and polypeptides processing.
Related to the previous point, in line 30, the sentence suggests that there are multiple cap structures at the 5’end. This should be clarified.
Response: we further clarify the 5’-cap structures in line 37 to 38.
When describing the two polypeptides (pp1a and pp1ab) the authors should indicate how they are generated.
Response: a brief description of polypeptides translation and nsps processing is added to line 42 to 46.
In line 41, there is an extra space after “ion channel and ..”
Response: it has been deleted.
In line 50, it should read “live attenuated vaccines”
Response: it has been revised.
In Line 52, introduction of Vero cells should be polypeptides accompanied by a description that defines it as a cell line incapable of secreting IFN but that still are expressing IFN receptors
Response: this point has been added to line 64 to 65.
In line 90, the author should provide a brief description explaining why the vaccines that have been commercially available in the USA were not as efficacious.
Response: we believe that viral replication in the intestine is the key to stimulate sufficient protection against highly virulent PEDV, similar to the previous observation for TGEV LAVs. The two licensed PEDV vaccines are viral vector-based subunit and inactivated PEDV vaccines that can’t replicate in the intestine. These points are discussed in line 107 to 113.
Line 110: are the authors referring PC22A-P120, PT-96 and KNU-141112 DEL/ORF3 as G2 strains. This should be fixed in the text.
Response: phylogenetically, all these three strains are classified are the G2 or highly virulent strains, compared with the G1a/ classical strains or G1b/S INDEL strains.
Table 1 is hard to follow. The authors should add footnotes for clarity and should be consistent in the nomenclature of the different mutants.
Response: There are some footnotes in the table. The mutations are ordered in by the location in the PEDV genome.
In line 124: ER should be referred as endoplasmic reticulum and later in the text should be referred as ER.
Response: this point has been revised.
In line 137, attenuated virus Zhejian08 has not been described in table 1.
Response: There is no genomic sequences for both virulent and attenuated Zhejiang08 in the original publication or at GenBank.
The text of this section: Mutations in Vero cells-attenuated G2 PEDV strains would benefit from separating paragraphs to help readers.
Response: This section has been divided into four paragraphs.
The text should include the specific abbreviations for the different amino acids
Response: We describe amino acid mutation into one-letter abbreviation form which is widely accepted by the science field. There are about ~20 amino acids that have been mentioned in this review manuscript. It would be very redundant to specify all of them in the text.
The following statement: “the higher the passages in Vero cells, the less replication of the virus is, leading to milder diseases and weaker immune responses in neonatal or weaned pigs [63,64]. One potential reason is the excessive adaptation of PEDV to the simian cell line resulted in accumulated mutations in the S protein” has already being discussed before. The authors should considered reorganizing the sections to avoid overlapping and redundancies.
Response: we have deleted the redundant sentences in the text.
The topic of recombination should be addressed when discussing viral evolution (line 79)
Response: Thank you for this suggestion. PEDV evolution is not a main topic in this review article, therefore, we did not mention it too much. We do briefly introduce PEDV recombination in lines 91-92, 99, 308, 326, 328 and 329.
Round 2
Reviewer 2 Report
The authors have addressed all the concerns raised during the first round of review.